# Cystic Fibrosis and Cancer: Unraveling the Complex Role of CFTR Gene in Cancer Susceptibility

**DOI:** 10.3390/cancers15174244

**Published:** 2023-08-24

**Authors:** Giuseppe Fabio Parisi, Maria Papale, Giulia Pecora, Novella Rotolo, Sara Manti, Giovanna Russo, Salvatore Leonardi

**Affiliations:** 1Pediatric Respiratory Unit, Department of Clinical and Experimental Medicine, San Marco Hospital, University of Catania, Viale Carlo Azeglio Ciampi sn, 95121 Catania, Italy; mariellapap@yahoo.it (M.P.); giupec87@hotmail.it (G.P.); rotolo@policlinico.unict.it (N.R.); leonardi@unict.it (S.L.); 2Pediatric Unit, Department of Human and Pediatric Pathology “Gaetano Barresi”, AOUP G. Martino, University of Messina, Via Consolare Valeria, 1, 98124 Messina, Italy; sara.manti@unime.it; 3Pediatric Hematology and Oncology Unit, Department of Clinical and Experimental Medicine, University of Catania, 95123 Catania, Italy; diberuss@unict.it

**Keywords:** cystic fibrosis, CFTR gene, cancer risk, life expectancy, genetic factors, colorectal cancer, pancreatic cancer, breast cancer, respiratory cancers, CFTR modulator therapies

## Abstract

**Simple Summary:**

Cystic fibrosis (CF) is a genetic condition that affects the lungs, digestion, and other body systems. People with CF have a higher chance of developing certain types of cancer. The reason for this is related to a gene called CFTR, which is altered in CF patients. This gene normally helps regulate the movement of substances in and out of cells. When it does not work properly, it can lead to changes in cells that make them more likely to become cancerous. The cancers most commonly associated with CF are colorectal, pancreatic, and respiratory cancers. By understanding how CFTR and cancer are connected, doctors can develop better ways to screen for and treat these cancers in people with CF. More research is needed to fully understand this link and improve care for CF patients.

**Abstract:**

Cystic fibrosis (CF) is a genetic disorder affecting multiple organs, primarily the lungs and digestive system. Over the years, advancements in medical care and treatments have significantly increased the life expectancy of individuals with CF. However, with this improved longevity, concerns about the potential risk of developing certain types of cancers have arisen. This narrative review aims to explore the relationship between CF, increased life expectancy, and the associated risk for cancers. We discuss the potential mechanisms underlying this risk, including chronic inflammation, immune system dysregulation, and genetic factors. Additionally, we review studies that have examined the incidence and types of cancers seen in CF patients, with a focus on gastrointestinal, breast, and respiratory malignancies. We also explore the impact of CFTR modulator therapies on cancer risk. In the gastrointestinal tract, CF patients have an elevated risk of developing colorectal cancer, pancreatic cancer, and possibly esophageal cancer. The underlying mechanisms contributing to these increased risks are not fully understood, but chronic inflammation, altered gut microbiota, and genetic factors are believed to play a role. Regular surveillance and colonoscopies are recommended for early detection and management of colorectal cancer in CF patients. Understanding the factors contributing to cancer development in CF patients is crucial for implementing appropriate surveillance strategies and improving long-term outcomes. Further research is needed to elucidate the molecular mechanisms involved and develop targeted interventions to mitigate cancer risk in individuals with CF.

## 1. Introduction

Cystic fibrosis (CF) is a complex genetic disorder that primarily affects the respiratory and digestive systems. It is caused by mutations in the cystic fibrosis transmembrane conductance regulator (CFTR) gene, resulting in dysfunctional CFTR protein [1,2]. CF’s hallmark is its impact on the production of thick, sticky mucus, which obstructs airways, leading to recurrent respiratory infections and impaired lung function. Additionally, the digestive tract’s secretions are affected, causing challenges in nutrient absorption and digestive processes [3].

While CF is well-known for its impact on lung function and digestive processes, the increase in life expectancy with the introduction of new highly effective CFTR modulator (HEMT) therapy and recent evidence suggests a potential association between CF and an increased risk of developing certain types of cancers [4,5,6,7,8,9]. In recent years, research efforts have focused on elucidating the underlying mechanisms and understanding the implications of this relationship. 

The genetic basis of CF lies in the CFTR gene, which regulates the flow of chloride ions across cell membranes. Mutations in the CFTR gene lead to impaired ion transport, resulting in the characteristic symptoms of CF. However, these mutations have also been implicated in various cellular processes that influence cancer development and progression [10,11]. Recent studies have focused on the potential impact of CFTR dysfunction on key pathways involved in carcinogenesis, such as cell proliferation, apoptosis, and DNA repair mechanisms [7,8,9]. Understanding these molecular mechanisms is crucial for unraveling the link between CF and cancer risk [12]. 

In addition to the genetic and molecular aspects, CF is characterized by chronic inflammation and dysregulated immune responses. The chronic inflammatory state in CF is primarily driven by the dysfunctional CFTR protein and is evident in both the respiratory and digestive systems [13]. This chronic inflammation can create a pro-tumorigenic microenvironment that promotes the initiation and progression of cancers [14]. Recent research has shed light on the role of inflammatory mediators, immune cells, and altered immune responses in the context of CF-related cancers [15,16]. Exploring the immunological aspects of CF and their influence on cancer development is essential for a comprehensive understanding of the disease.

Several specific types of cancer have been associated with CF, including colorectal, pancreatic, breast, and respiratory malignancies [7,8,9]. Epidemiological studies have consistently shown an increased incidence of some of these cancers in individuals with CF compared to the general population [17]. The specific mechanisms underlying the increased risk remain the subject of ongoing research. Factors such as chronic inflammation, altered immune response, gut microbiota dysbiosis, and CFTR dysfunction likely contribute to the development of these cancers in individuals with CF [18].

The implications of these findings extend beyond understanding the association between CF and cancer risk. They have significant clinical implications for the management of individuals with CF, including screening, surveillance, and treatment strategies. Tailored screening protocols are necessary to facilitate early detection, while surveillance for specific cancers should be incorporated into routine CF care. Furthermore, collaborations between CF care teams and oncology specialists are vital for providing comprehensive care to individuals with CF-related cancers [19].

This comprehensive review aims to accomplish the following objectives:Synthesize Current Knowledge: Summarize and consolidate the existing literature on the relationship between CF, the CFTR gene, and cancer susceptibility.Examine Specific Cancer Associations: Investigate the associations between CFTR gene mutations and the risk of specific cancer types, including pancreatic, respiratory, colorectal, breast, liver, esophageal, and gastric cancers.Explore Underlying Mechanisms: Explore the molecular and cellular mechanisms by which CFTR gene mutations may influence cancer susceptibility, encompassing factors such as chronic inflammation, impaired DNA repair, hormonal imbalances, and other cellular processes.Highlight Emerging Research: Highlight recent advancements and emerging research that shed light on the complex interplay between CF, CFTR gene mutations, and cancer development.Identify Knowledge Gaps: Identify gaps in the current understanding of the CFTR–cancer relationship, pinpointing areas that require further research and investigation.Clinical Implications: Discuss the potential clinical implications of the CFTR–cancer connection, including its impact on cancer surveillance, early detection, and potential therapeutic interventions.Inform Future Research Directions: Propose future research directions and methodologies that could elucidate the intricate mechanisms underlying the association between CFTR gene mutations and cancer susceptibility.

## 2. Methods

This comprehensive review of the association between CF and cancers was conducted using a narrative approach to gather and analyze relevant literature over the past decade. A comprehensive search of electronic databases, including PubMed, Scopus, and Web of Science, was conducted to identify relevant articles published in English. The search strategy incorporated keywords related to cystic fibrosis, cancer, malignancy, CFTR, and associated terms. The search was limited to articles published from 1990. 

The initial search yielded a large number of articles (>500). Duplicate articles were removed, and titles and abstracts were screened for relevance. Full-text articles of potentially relevant studies were retrieved and assessed for eligibility. The inclusion criteria encompassed studies that investigated the association between CF and cancers. Review articles, original research papers, and case reports were considered.

Data from selected articles were extracted systematically. The following information was collected: author(s), publication year, study design, study population, cancer type, sample size, methods used for data collection, and key findings related to the association between CF and cancers. Additionally, data on the genetic, molecular, and immunological aspects of CF that contribute to cancer risk were extracted.

The extracted data were synthesized to provide a comprehensive overview of recent findings. The information was organized thematically, focusing on the genetic, molecular, and immunological aspects of CF, as well as specific types of cancer associated with CF, such as colorectal, pancreatic, breast, and respiratory malignancies. Key findings were summarized, and relevant concepts were discussed in detail.

The included studies were critically evaluated for their methodological quality and potential biases. Any limitations or gaps in the current literature were identified and discussed. The strengths and weaknesses of the studies were taken into account during the interpretation of the results.

The information gathered from the data synthesis and critical analysis was used to develop the manuscript. The review was structured to provide a comprehensive overview of recent findings regarding the association between CF and cancers. The introduction, methods, results, and discussion sections were written, highlighting the key aspects and implications of the findings.

## 3. Role of CFTR in Cancers

The role of CFTR in cancer is an intriguing area of investigation that has gained substantial interest in recent years. Initially recognized for its involvement in CF, CFTR has emerged as a potential player in cancer development and progression [20].

One of the key aspects of CFTR’s role in cancer is its influence on ion transport and cellular homeostasis. CFTR acts as a chloride channel, regulating the movement of chloride ions and water across cell membranes. Dysregulation of CFTR can disrupt ion transport, leading to altered cellular homeostasis. This disruption has been associated with changes in cellular pH regulation and metabolism, both of which have significant implications for cancer cell growth and survival [21]. For example, CFTR dysfunction may disturb the balance of intracellular chloride and bicarbonate ions, affecting pH levels and influencing critical metabolic processes in cancer cells [22]. Furthermore, CFTR has been implicated in the regulation of epithelial-mesenchymal transition (EMT), a fundamental process involved in cancer metastasis. EMT is characterized by the loss of epithelial cell characteristics and the acquisition of a more mesenchymal-like phenotype, enabling cells to invade surrounding tissues and metastasize to distant sites. CFTR has been shown to modulate EMT through various mechanisms. CFTR dysfunction can lead to alterations in ion transport, calcium signaling, and the activity of signaling pathways such as transforming growth factor-beta (TGF-β) and Wnt/β-catenin, all of which play crucial roles in EMT regulation [23]. These changes in EMT-related pathways can contribute to increased invasiveness and metastatic potential of cancer cells.

Inflammation is a well-established driver of cancer development, and CFTR dysfunction has been associated with elevated levels of inflammation in various tissues. CFTR mutations can result in increased production of pro-inflammatory cytokines and chemokines, creating a pro-tumorigenic microenvironment [24]. Moreover, CFTR dysfunction may impact immune responses, influencing the infiltration and activation of immune cells within the tumor microenvironment. The dysregulated immune response in the presence of CFTR dysfunction may further contribute to cancer progression [25].

CFTR also appears to be involved in cellular proliferation and survival pathways. CFTR dysfunction can modulate the activity of signaling pathways such as phosphoinositide 3-kinase (PI3K)/Akt and mitogen-activated protein kinase (MAPK), which are crucial for cell growth and survival [23,26]. Additionally, CFTR may interact with other proteins involved in cell cycle regulation and apoptosis, influencing the behavior and survival of cancer cells [27]. These molecular interactions and alterations in signaling pathways can contribute to uncontrolled cellular proliferation and resistance to cell death mechanisms. 

Furthermore, CFTR has been implicated in drug resistance in certain cancers [28]. The activity of CFTR can affect the response of cancer cells to chemotherapeutic agents by influencing drug uptake, efflux, and intracellular concentration. CFTR-mediated drug resistance can impact the effectiveness of cancer treatment and pose challenges in achieving successful outcomes.

In summary, CFTR plays a multifaceted role in cancer, influencing various aspects of tumor biology, including ion transport, EMT, inflammation, cellular proliferation, and drug response (Figure 1). Dysregulation of CFTR can have profound effects on cancer development and progression, further elucidating the molecular mechanisms underlying CF.

## 4. CF and Gastrointestinal Cancers

Gastrointestinal cancers encompass a range of malignancies affecting the digestive tract, including the esophagus, stomach, small intestine, colon, rectum, pancreas, and liver. Understanding the link between CF and gastrointestinal cancers is important for improving patient care, implementing appropriate surveillance and screening measures, and identifying potential therapeutic interventions.

### 4.1. Esophageal Cancer

Esophageal cancer is a relatively rare but aggressive malignancy that poses significant challenges to patients and healthcare providers. While the exact molecular mechanisms linking CF and esophageal cancers are not yet fully understood, several factors have been proposed to contribute to this association [7].

One potential mechanism is chronic inflammation resulting from CFTR dysfunction. The thickened mucus and impaired clearance in the respiratory and digestive systems of CF patients create an environment conducive to chronic inflammation. Chronic inflammation is known to play a key role in carcinogenesis, and it has been suggested that long-term inflammation in the esophagus may increase the risk of developing esophageal cancer in individuals with CF [29].

Additionally, CFTR mutations may impact the composition of the esophageal microbiota. Dysbiosis, an imbalance of bacterial species in the esophagus, has been associated with an increased risk of esophageal diseases, including esophageal cancer. CF patients may have altered esophageal microbiota due to the effects of CFTR dysfunction, and this dysbiosis could potentially contribute to the development of esophageal malignancies [30].

Moreover, CF patients often face nutritional challenges due to malabsorption and malnutrition. These nutritional issues may lead to deficiencies in key vitamins and minerals, which are essential for maintaining cellular health and DNA repair mechanisms. Such deficiencies could increase the susceptibility to cellular damage and the risk of developing esophageal cancers [31].

Furthermore, recent studies have implicated specific molecular pathways in the association between CFTR dysfunction and esophageal cancer development. For example, it has been suggested that CFTR mutations may lead to alterations in calcium signaling pathways, which play a critical role in cell proliferation, differentiation, and apoptosis. Dysregulation of calcium signaling can contribute to uncontrolled cell growth and the development of cancer [32].

Another potential molecular mechanism involves the disruption of epithelial cell homeostasis in the esophagus. CFTR dysfunction may impair the transport of bicarbonate ions, which are important for maintaining the proper pH balance in the esophageal epithelium. This disruption can lead to cellular stress, DNA damage, and increased susceptibility to carcinogenesis [33,34].

Finally, adults with CF have a higher risk of developing Barret’s esophagus, which is a precursor for esophageal cancer [35].

Despite these potential associations and molecular mechanisms, the exact link between CF and esophageal cancers remains an active area of research. The relative risk of esophageal cancer in patients with CF is not well-established due to limited available data. As of now, there is a lack of consensus on the specific relative risk values for esophageal cancer in CF patients. The rarity of esophageal cancer in CF patients and the complexity of its underlying mechanisms make it challenging to draw definitive conclusions. Further studies are needed to elucidate the precise molecular pathways connecting CFTR dysfunction and the development of esophageal malignancies.

### 4.2. Gastric Cancer

Gastric cancer, also known as stomach cancer, is a malignant tumor that develops in the stomach lining. CFTR dysfunction in the stomach can lead to the accumulation of thick mucus, impairing mucociliary clearance. The retained mucus creates a favorable environment for bacterial colonization, resulting in chronic gastritis and inflammation. Chronic inflammation, characterized by the release of pro-inflammatory cytokines, chemokines, and growth factors, can promote genetic mutations, stimulate cellular proliferation, and enhance angiogenesis, ultimately contributing to the development and progression of gastric cancer [36,37].

CFTR plays a role in regulating chloride and bicarbonate ion transport, which impacts gastric acid secretion. CFTR dysfunction can lead to altered gastric acid production and pH levels. Reduced gastric acid secretion may increase the risk of gastric cancer by impairing microbial defense mechanisms and promoting the growth of Helicobacter pylori, a bacterium implicated in gastric cancer development. Moreover, altered gastric pH can affect the digestion and absorption of dietary factors that may modulate gastric carcinogenesis [38,39].

In addition to CFTR mutations, CF-related genetic variations may contribute to the increased risk of gastric cancer. Genome-wide association studies have identified certain genetic variants associated with both CF and gastric cancer susceptibility. These genetic variations may affect immune response, DNA repair mechanisms, or other processes involved in gastric carcinogenesis, highlighting potential shared genetic pathways between CF and gastric cancer [40,41].

Various other molecular pathways have been implicated in the association between CFTR dysfunction and gastric cancer development. For instance, CFTR dysfunction may lead to altered calcium signaling, affecting cell proliferation, differentiation, and apoptosis, which are critical processes in gastric carcinogenesis. Disruption of epithelial cell homeostasis, impaired bicarbonate ion transport, and subsequent cellular stress and DNA damage may also contribute to the development of gastric cancer in CF [42,43].

Therefore, the association between CF, CFTR, and gastric cancer involves complex molecular mechanisms. Chronic inflammation, altered gastric acid secretion, CF-related malnutrition, genetic variations, and disrupted cellular pathways collectively contribute to the increased risk of gastric cancer in individuals with CF. 

Similar to esophageal cancer, the relative risk of gastric cancer in patients with CF is not well-defined. The available data on the association between CF and gastric cancer are limited, making it challenging to estimate precise relative risk values for this specific cancer type.

Further research is needed to fully elucidate these mechanisms and their interplay in the development and progression of gastric cancer in CF patients. 

### 4.3. Pancreatic Cancer

Pancreatic cancer is a devastating disease characterized by its aggressiveness and poor prognosis. CF patients have an increased relative risk of developing pancreatic cancer. Studies have reported relative risk values ranging from 5 to 10 times higher in CF patients compared to the general population [17,19,44,45]. CFTR dysfunction caused by mutations in the CFTR gene leads to abnormal ion transport across epithelial cells, including those lining the pancreatic ducts. The resulting impaired CFTR function leads to altered fluid secretion and increased viscosity of pancreatic secretions, ultimately leading to ductal obstruction. The accumulation of thickened secretions creates a microenvironment conducive to inflammation, fibrosis, and cellular damage, potentially predisposing individuals with CF to pancreatic cancer [44,45].

CFTR dysfunction and pancreatic duct obstruction trigger chronic inflammation in the pancreas. Inflammatory processes involve the release of pro-inflammatory cytokines, chemokines, and reactive oxygen species, leading to cellular damage and genetic mutations. Prolonged inflammation can induce DNA damage, dysregulate cellular signaling pathways, and disturb cell growth and survival mechanisms, all of which are implicated in pancreatic cancer development [46].

CF-related pancreatic insufficiency often coexists with bile duct abnormalities and impaired bile flow. These conditions can result in increased exposure of pancreatic tissue to bile acids, digestive enzymes, and duodenal reflux. The duodenal refluxate, consisting of bile acids and other duodenal contents, can cause cellular injury, inflammation, and oxidative stress in the pancreas. Sustained exposure to these damaging factors may contribute to the initiation and progression of pancreatic cancer [47,48].

In addition to CFTR mutations, CF-related genetic factors have been implicated in pancreatic cancer development. Genome-wide association studies have identified specific genetic variants associated with both CF and pancreatic cancer susceptibility. These variants may affect immune response, cellular metabolism, or other pathways involved in pancreatic carcinogenesis. Investigating these shared genetic factors can provide valuable insights into the molecular mechanisms connecting CF and pancreatic cancer [49,50].

Growing evidence suggests that alterations in the gut microbiota, known as dysbiosis, may play a role in pancreatic cancer development. CF-related pancreatic insufficiency, altered bile flow, and impaired digestive processes can disrupt the gut microbial ecosystem. Dysbiosis in CF patients may result in the production of harmful metabolites, chronic inflammation, and perturbation of the host–microbiota interaction, which may contribute to pancreatic carcinogenesis [51,52].

In conclusion, the association between CF, CFTR, and pancreatic cancer involves intricate molecular mechanisms. CFTR dysfunction, pancreatic duct obstruction, chronic inflammation, altered bile flow, CFTR-related genetic factors, impaired nutrient absorption, and microbiota dysbiosis collectively contribute to the increased risk of pancreatic cancer in individuals with CF. Further research is necessary to fully elucidate these molecular mechanisms and their interplay in the development and progression of pancreatic cancer in CF patients.

### 4.4. Liver Cancer

CFTR dysfunction resulting from CF-associated mutations disrupts chloride and bicarbonate transport, leading to impaired bile secretion and altered bile composition. This disturbance in bile flow can cause cholestasis and subsequent hepatic fibrosis [53]. Prolonged fibrotic changes in the liver microenvironment create a pro-inflammatory milieu and promote cellular proliferation, thereby increasing the risk of hepatocellular carcinoma (HCC). Studies have shown that CF patients with liver cirrhosis have an increased risk of developing HCC [54,55].

CFTR dysfunction contributes to chronic inflammation and oxidative stress in the liver. Impaired CFTR function leads to the accumulation of bile acids, which can induce oxidative damage and activate inflammatory pathways. Chronic inflammation and oxidative stress create a favorable environment for the development of hepatic cancer by promoting DNA damage, genomic instability, and cellular proliferation. Studies have demonstrated increased levels of pro-inflammatory markers and oxidative stress in CF-related liver disease [56,57].

CFTR has been shown to play a role in liver regeneration. During liver injury, CFTR expression is upregulated, suggesting its involvement in the regenerative process. CFTR-deficient mice exhibit impaired liver regeneration, suggesting that altered CFTR expression and function may disrupt the regenerative capacity of liver cells. Impaired liver regeneration can contribute to the development of hepatic cancer [58,59].

In addition to CFTR mutations, other CF-related genetic factors have been associated with an increased risk of hepatic cancer. Genetic variations in CFTR modifier genes, such as the Solute Carrier Organic Anion Transporter (SLCO) family, have been implicated in hepatocarcinogenesis. These variations may affect drug metabolism, transport, and cellular pathways involved in liver cancer development. Studies have identified associations between CFTR-related genetic variations and increased susceptibility to liver cancer in CF patients [60,61].

CF patients may have a slightly elevated risk of developing liver cancer, although the relative risk values vary across studies. Relative risk values around 1.5 to 2.0 have been suggested [17,19].

In summary, there has been growing evidence supporting a link between CF, CFTR dysfunction, and the development of hepatic cancer. The molecular mechanisms underlying this association involve CFTR dysfunction-related hepatic fibrosis, chronic inflammation, oxidative stress, impaired liver regeneration, CFTR-related genetic factors, and nutritional deficiencies. 

### 4.5. Intestinal Cancers

#### 4.5.1. Colorectal Cancer

Colorectal cancer (CRC) is a malignant neoplasm that arises from the epithelial cells lining the colon or rectum. Patients with CF are at a 6-fold higher risk for CRC [19,37,62]. The mechanisms underlying this association are not yet fully understood, but several factors have been implicated.

CFTR dysfunction in the intestinal epithelium leads to persistent inflammation and oxidative stress. The impaired CFTR function affects ion transport, mucus clearance, and the integrity of the intestinal barrier [63,64]. These disruptions create an environment conducive to chronic inflammation and oxidative stress, which can promote the development of CRC. Inflammation and oxidative stress induce DNA damage, genomic instability, and cellular proliferation, key factors in carcinogenesis.

Recent studies have highlighted the role of pro-inflammatory cytokines, such as interleukin-6 (IL-6) and tumor necrosis factor-alpha (TNF-α), in CRC development. Increased expression of these cytokines has been observed in CF patients, indicating a potential link between CF-associated inflammation and CRC [65,66]. Additionally, oxidative stress resulting from impaired CFTR function can lead to the accumulation of reactive oxygen species (ROS), causing DNA damage and favoring the initiation and progression of CRC [67,68].

CF patients often exhibit dysbiosis, an imbalance in the composition and function of the gut microbiota. Dysbiosis in CF is characterized by a reduction in beneficial bacteria, such as Bifidobacterium and Lactobacillus species, and an increase in potentially harmful bacteria, including Enterobacteriaceae and Pseudomonas aeruginosa. Dysbiosis can contribute to inflammation, impaired intestinal barrier function, and increased susceptibility to CRC. The specific dysbiosis patterns associated with CRC in CF patients warrant further investigation [69,70].

Recent studies have highlighted the potential role of specific bacterial species in CRC development. For example, Fusobacterium nucleatum, a common member of the gut microbiota, has been associated with CRC progression by promoting inflammation and impairing immune surveillance [71,72]. In CF patients, dysbiosis and altered microbial composition may create a microenvironment conducive to the growth of pathogenic bacteria, further contributing to the development of CRC.

In addition to CFTR mutations, CF-related genetic factors may influence the risk of CRC development in CF patients. Modifier genes that interact with CFTR, such as those involved in inflammation, immune response, and cellular proliferation, may play a role in CRC susceptibility. Variations in these genes can modify the disease phenotype and influence the development of CRC in CF patients [73,74].

Recent studies have identified genetic polymorphisms associated with both CF and CRC, suggesting a potential genetic link between the two conditions. For example, the TNF-α gene polymorphism has been implicated in both CF and CRC susceptibility [75]. These genetic factors may modulate the inflammatory response, alter immune cell function, and contribute to the development of CRC in CF patients.

Current guidelines recommend CRC surveillance for CF patients starting at the age of 40 or 10 years before the youngest affected relative’s diagnosis (whichever comes first). The surveillance typically involves periodic colonoscopies with the aim of detecting precancerous polyps or early-stage CRC. Additionally, individuals with CF who present with concerning symptoms such as unexplained gastrointestinal bleeding or persistent change in bowel habits should undergo timely evaluation [76,77].

#### 4.5.2. Small Bowel Adenocarcinoma

Small bowel adenocarcinoma (SBA) is a rare but aggressive form of intestinal cancer that can occur in CF patients. The underlying mechanisms linking CF and SBA are not yet fully elucidated, but several factors may contribute to its development.

Several tumor suppressor genes have been implicated in SBA development, including TP53, APC, and SMAD4. TP53, commonly known as the “guardian of the genome”, plays a crucial role in DNA repair and cell cycle regulation. CFTR dysfunction could potentially affect TP53 function, compromising its ability to suppress tumor formation and progression in the small intestine [78,79]. Further studies are needed to elucidate the specific molecular interactions between CFTR and tumor suppressor genes in the context of SBA development.

## 5. Breast Cancer

The prevalence of breast cancer in CF patients is generally not higher than that in the general population. However, with advancements in CF treatments, individuals with CF are living longer, and there is a growing population of women with CF reaching the age at which breast cancer becomes more common [9].

Estrogens play a significant role in both the pathophysiology of CF and breast cancer. However, their effects on these two conditions are distinct and require separate considerations. In the context of CF, estrogen has been shown to exert beneficial effects on lung function and disease progression. CF is characterized by abnormal ion transport due to mutations in the cystic fibrosis transmembrane conductance regulator (CFTR) gene. Estrogen has been found to enhance CFTR function and increase chloride secretion in the airways, leading to improved mucus clearance and lung function [80,81]. Estrogen’s protective effects on lung function may be attributed to its ability to stimulate CFTR expression and activity through various signaling pathways, including cyclic adenosine monophosphate (cAMP)-dependent mechanisms [82].

In contrast, estrogen plays a complex role in the pathophysiology of breast cancer. Estrogen receptor (ER) signaling is known to promote the growth and proliferation of breast cancer cells. In hormone receptor-positive breast cancers, estrogen binds to ERs, leading to the activation of downstream signaling pathways that drive tumor cell growth and survival. Estrogen also promotes angiogenesis, the formation of new blood vessels, which is crucial for tumor growth and metastasis [83,84,85].

It is worth noting that the use of hormone replacement therapy (HRT) in CF patients needs careful consideration. While HRT may have potential benefits for improving lung function and bone health in postmenopausal CF women, it also carries potential risks, including the promotion of hormone-sensitive cancers such as breast cancer [86]. The decision to use HRT should be made on an individual basis, taking into account the patient’s overall health status and the potential benefits and risks.

While there is limited data on breast cancer risk specifically in CF, it is important to consider appropriate screening strategies for CF patients. Following general breast cancer screening guidelines is recommended, including regular clinical breast exams, mammography, and breast self-examinations (American Cancer Society). Screening mammograms typically begin at age 40 and continue annually for women at average risk of breast cancer [87].

CF-related factors may present challenges in breast cancer screening and management. CF-related lung disease can make it difficult for patients to undergo mammography due to positioning and breathing difficulties. In such cases, alternative imaging modalities such as breast ultrasound or magnetic resonance imaging (MRI) may be considered [88]. Collaborating with healthcare providers experienced in managing breast cancer screening in individuals with CF can help develop appropriate and effective screening strategies.

Breast cancer screening and management should be integrated into the comprehensive care of CF patients. A multidisciplinary approach involving CF specialists, oncologists, genetic counselors, and other healthcare providers is crucial to address the unique needs and challenges of CF patients regarding breast cancer. Close coordination and communication among the different healthcare professionals involved are important to ensure comprehensive and coordinated care.

Psychosocial support should also be provided throughout the breast cancer screening and management process. CF patients may already face significant physical and emotional burdens related to their condition, and breast cancer screening and potential diagnosis can add additional emotional challenges. Counseling services, support groups, and resources can help CF patients navigate the emotional aspects of breast cancer screening and potential diagnosis.

## 6. Lung Cancers

Lung cancer is one of the most common malignancies worldwide, and individuals with CF have an increased, although still not quantifiable, risk of developing certain types of lung cancers. The role of the CFTR in CF and its relationship to lung cancer have been the focus of scientific investigation. Epidemiologic data reveal that individuals with CF have an increased risk of developing certain types of lung cancers compared to the general population [89,90]. One notable subtype of lung cancer that is more prevalent in CF patients is bronchial gland carcinoma, which arises from the mucous glands in the airways. While bronchial gland carcinomas are relatively rare in the general population, they occur more frequently in CF patients [91,92]. The specific mechanisms underlying this increased susceptibility to bronchial gland carcinomas in CF are still being investigated.

In addition to bronchial gland carcinomas, CF patients also have a heightened risk of other lung malignancies, such as squamous cell carcinoma and adenocarcinoma [17,93]. These types of lung cancers are commonly associated with smoking, and CF patients who smoke face an even higher risk of developing lung cancer compared to non-smoking CF patients. Therefore, smoking cessation is strongly encouraged in CF patients to reduce the risk of lung cancer and other smoking-related health complications [94,95].

Several factors contribute to the increased risk of lung cancer in CF patients. Chronic inflammation and tissue damage in the lungs, often caused by chronic bacterial infections like Pseudomonas aeruginosa, play a crucial role. These infections lead to persistent inflammation and oxidative stress, creating an environment that promotes tumor development. Moreover, the genetic mutations in the CFTR gene, resulting in CFTR dysfunction, may also contribute to an altered cellular environment that favors the development of lung cancer [90,93].

It is important to note that despite the increased risk, the overall incidence of lung cancer in CF patients remains relatively low compared to the general population. The improved survival and enhanced quality of life in CF patients due to advancements in CF treatments and therapies may contribute to the increased likelihood of reaching an age where lung cancer becomes more common. Regular monitoring and screening for lung cancer are crucial in CF patients, particularly those with additional risk factors like smoking, as early detection can lead to improved outcomes.

## 7. Other Emerging Cancers

Although rare, cases of thyroid tumors, melanomas, ovarian cancers, and brain tumors have been reported [17,62,90,96].

Other emerging cancers in CF, including bone cancer, soft tissue sarcoma, bladder cancer, prostate cancer, and uterine cancer, have limited data available [91]. The potential impact of CFTR dysfunction on these cancers requires more comprehensive studies to establish a clearer association.

It is important to note that the epidemiologic data for emerging cancers in CF are limited, and further research is necessary to better understand the prevalence and molecular mechanisms involved. Advancements in research will contribute to improved screening, prevention, and management strategies for cancer in patients with CF.

## 8. Highly Effective CFTR Modulator Therapy (HEMT) and Cancers

CFTR modulator therapy has revolutionized the treatment landscape for CF by targeting the underlying defect in the CFTR gene [97,98]. However, little is known about the possible long-term effects and their potential impact on cancer risk. At present, the available data on the long-term effects of modulator therapies on cancer risk in CF patients are limited due to the relatively recent introduction of these drugs and the need for long-term follow-up studies. However, based on the current knowledge and studies conducted thus far, there is no conclusive evidence to suggest that modulator therapy increases the overall risk of cancer in CF patients [99,100,101,102,103,104].

## 9. Conclusions

The role of the CFTR gene in the development and progression of cancers in patients with CF is an emerging area of research (Table 1). Although CF primarily affects the respiratory and gastrointestinal systems, evidence suggests that CFTR gene mutations may also increase the risk of specific cancers in CF patients.

In the gastrointestinal tract, CF patients have an elevated risk of developing colorectal cancer, pancreatic cancer, and possibly esophageal cancer. The underlying mechanisms contributing to these increased risks are not fully understood, but chronic inflammation, altered gut microbiota, and genetic factors are believed to play a role. Regular surveillance and colonoscopies are recommended for early detection and management of colorectal cancer in CF patients.

The advent of CFTR modulator therapies has significantly improved the clinical outcomes of CF patients by correcting CFTR dysfunction. However, concerns have been raised about the potential long-term effects of CFTR modulator therapy on cancer risk. Further research is needed to clarify the relationship between CFTR modulator therapy and cancer development in CF patients.

In conclusion, the CFTR gene, responsible for the pathogenesis of CF, may also play a role in the development and progression of certain cancers in CF patients. Understanding the molecular mechanisms underlying these associations and identifying effective surveillance and management strategies are crucial for optimizing the care of CF patients and mitigating cancer risks. Continued research in this field will contribute to the development of personalized approaches to cancer prevention, screening, and treatment in individuals with CF. The possible clinical implications of these observations are profound, as they pave the way for enhanced cancer surveillance, tailored early detection strategies, and potential targeted therapies, ensuring comprehensive care for individuals with both CF and cancer predisposition.

## Figures and Tables

**Figure 1 cancers-15-04244-f001:**
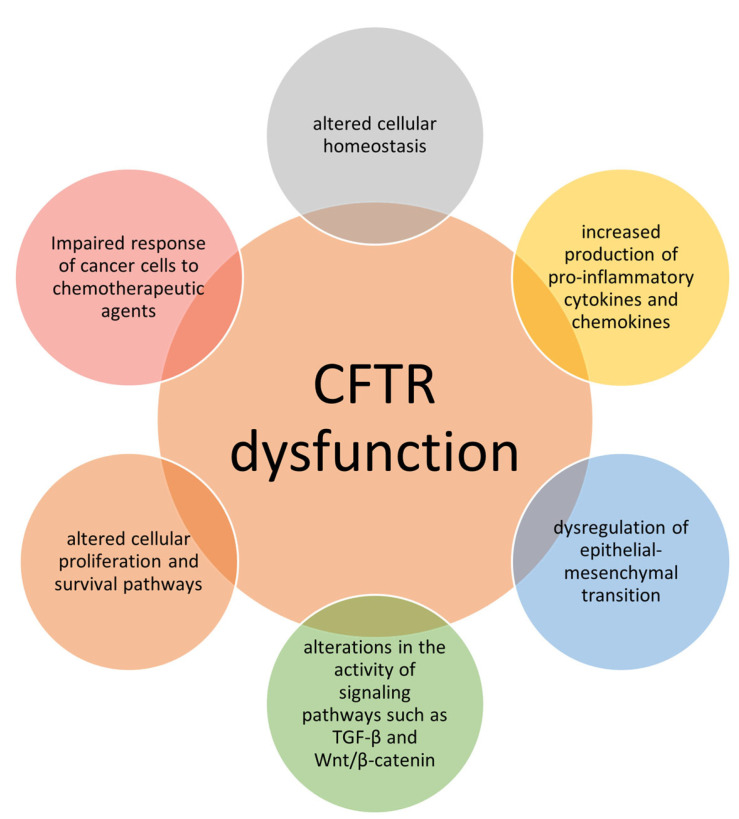
CFTR dysfunction and mechanisms related to predisposition to cancers. CFTR dysfunction in CF triggers chronic inflammation, impaired DNA repair, and hormonal imbalances. These mechanisms collectively predispose individuals to various cancers, highlighting the intricate interplay between CF and cancer susceptibility.

**Table 1 cancers-15-04244-t001:** Main cancers associated with cystic fibrosis.

Cancer Type	Molecular Mechanisms	Relative Risk
Esophageal cancer	Chronic inflammation Altered composition of esophageal microbiota Alterations in calcium signaling pathways Disruption of epithelial cell homeostasis Higher risk of developing Barret’s esophagus	not well-established
Gastric cancer	Chronic inflammation Altered gastric acid production and pH levels Alterations in calcium signaling pathways Disruption of epithelial cell homeostasis	not well-established
Pancreatic cancer	Chronic inflammation Altered bile flow Oxidative stress	5–10
Liver cancer	Chronic inflammation Altered bile flow Impaired liver regeneration Genetic variations in modifier genes, such as the Solute Carrier Organic Anion Transporter (SLCO) family	1.5–2
Intestinal cancers	Chronic inflammation Oxidative stress Altered composition of intestinal microbiota Genetic polymorphisms Implications of tumor suppressor genes	6
Breast cancer	Hormonal imbalances, such as increased estrogen levels	not well-established
Lung cancer	Chronic inflammation Altered mucociliary clearance	not well-established

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
