# Peer review of "Cystic Fibrosis and Cancer: Unraveling the Complex Role of CFTR Gene in Cancer Susceptibility"

_cancers, 2023, doi:10.3390/cancers15174244_

Round 1

Reviewer 1 Report

Strengths:

1. This is an interesting review that may be useful to those treating CF patients. 

2. The strategy of using a narrative approach based on a list of possible cancers is different and potentially useful to those with limited time. 

Concerns:

1. The references include many from more than 20 years ago. So, the comment in methods about limiting the survey to  more recent than  2010 needs to be revised. 

2. What needs to be added is data in each section on the relative risk of  the different types of cancer in CF patients. A summary Table with these data would make the review more useful.

Author Response

Strengths:

  1. This is an interesting review that may be useful to those treating CF patients.

Answer: Thank you.

  1. The strategy of using a narrative approach based on a list of possible cancers is different and potentially useful to those with limited time.

Answer: Thank you.

Concerns:

  1. The references include many from more than 20 years ago. So, the comment in methods about limiting the survey to more recent than 2010 needs to be revised.

Answer: Thank you. As you suggested, we modified the text in the methods section. 

  1. What needs to be added is data in each section on the relative risk of the different types of cancer in CF patients. A summary Table with these data would make the review more useful.

Answer: Thank you. As you requested, we added some information in the text regarding the relative risk and another column in table 1 summarizing the results.  

Reviewer 2 Report

The manuscript is quite well written. The topic is interesting. I have some suggestions:

1) Abstract.  Understanding the factors contributing to cancer development in CF patients is crucial for implementing appropriate surveillance strategies and improving long-term outcomes. Further research is needed to elucidate the molecular mechanisms involved and de-velop targeted interventions to mitigate cancer risk in individuals with CF. Please, improve the conclusions and summarise the most important observation of the review.

2) 1. Introduction L41-43. Cystic fibrosis (CF) is a complex genetic disorder that primarily affects the respiratory  and digestive systems. It is caused by mutations in the cystic fibrosis transmembrane  conductance regulator (CFTR) gene, resulting in dysfunctional CFTR protein [1, 2]. Could you please improve the introduction on Cystic fibrosis. I suggest some references:

a- Cancers 202315, 989. https://doi.org/10.3390/cancers15030989. Lower Expression of CFTR Is Associated with Higher Mortality in a Meta-Analysis of Individuals with Colorectal Cancer. 

b- Respir Med. 2021;189:106623. doi: 10.1016/j.rmed.2021.106623. Combined use of rheology and portable low-field NMR in cystic fibrosis patients.

c- International Journal of Molecular Sciences. 2020; 21(8):2891. https://doi.org/10.3390/ijms21082891. Cystic Fibrosis, CFTR, and Colorectal Cancer. 

3) Introduction. L82-85.This comprehensive review aims to provide an overview of recent findings regarding  the association between cystic fibrosis and cancers, highlighting the genetic, molecular,  and immunological aspects of CF that contribute to cancer risk. By deepening our under- standing of this association, we can develop targeted interventions and improve clinical  outcomes for individuals with cystic fibrosis and associated cancers. Please improve the description of study aim. 

4) Figure 1. CFTR dysfunction and mechanisms related to predisposition to cancers. Nice figure! Please, improve the legend. 

5) 3. Role of CFTR in cancers. Please, add a table to summarise the results of most important sudies.

6) L522-527.  In conclusion, the CFTR gene, responsible for the pathogenesis of CF, may also play a  role in the development and progression of certain cancers in CF patients.  Understanding the molecular mechanisms underlying these associations and identifying effective surveillance and management strategies are crucial for optimizing the care of  CF patients and mitigating cancer risks. Continued research in this field will contribute  to the development of personalized approaches to cancer prevention, screening, and  treatment in individuals with CF. Please, underline the possible clinical implications of these observations.

Author Response

The manuscript is quite well written. The topic is interesting. I have some suggestions:

1) Abstract.  Understanding the factors contributing to cancer development in CF patients is crucial for implementing appropriate surveillance strategies and improving long-term outcomes. Further research is needed to elucidate the molecular mechanisms involved and de-velop targeted interventions to mitigate cancer risk in individuals with CF. Please, improve the conclusions and summarise the most important observation of the review.

Answer: Thank you. As you requested, we expanded the abstract section.

2) 1. Introduction L41-43. Cystic fibrosis (CF) is a complex genetic disorder that primarily affects the respiratory  and digestive systems. It is caused by mutations in the cystic fibrosis transmembrane  conductance regulator (CFTR) gene, resulting in dysfunctional CFTR protein [1, 2]. Could you please improve the introduction on Cystic fibrosis. I suggest some references:

a- Cancers 2023, 15, 989. https://doi.org/10.3390/cancers15030989. Lower Expression of CFTR Is Associated with Higher Mortality in a Meta-Analysis of Individuals with Colorectal Cancer.

b- Respir Med. 2021;189:106623. doi: 10.1016/j.rmed.2021.106623. Combined use of rheology and portable low-field NMR in cystic fibrosis patients.

c- International Journal of Molecular Sciences. 2020; 21(8):2891. https://doi.org/10.3390/ijms21082891. Cystic Fibrosis, CFTR, and Colorectal Cancer.

Answer: Thank you. As you requested, we improved the introduction adding the references you mentioned.

3) Introduction. L82-85.This comprehensive review aims to provide an overview of recent findings regarding  the association between cystic fibrosis and cancers, highlighting the genetic, molecular,  and immunological aspects of CF that contribute to cancer risk. By deepening our under- standing of this association, we can develop targeted interventions and improve clinical  outcomes for individuals with cystic fibrosis and associated cancers. Please improve the description of study aim.

Answer: Thank you. As you requested, we improved the description of study aim.  

4) Figure 1. CFTR dysfunction and mechanisms related to predisposition to cancers. Nice figure! Please, improve the legend.

Answer: Thank you. As you requested, we improved the legend.

5) 3. Role of CFTR in cancers. Please, add a table to summarise the results of most important sudies.

Answer: Dear reviewer, thank you for your suggestion.  Chapter 3 is already summarized in Figure 1. If we inserted another table, the contents would be redundant. We prefer to leave Figure 1 rather than add another table to make the article graphically lighter. Thank you for your understanding.

6) L522-527.  In conclusion, the CFTR gene, responsible for the pathogenesis of CF, may also play a  role in the development and progression of certain cancers in CF patients.  Understanding the molecular mechanisms underlying these associations and identifying effective surveillance and management strategies are crucial for optimizing the care of  CF patients and mitigating cancer risks. Continued research in this field will contribute  to the development of personalized approaches to cancer prevention, screening, and  treatment in individuals with CF. Please, underline the possible clinical implications of these observations.

Answer: Thank you. As you requested, we added the possible clinical implications.

Reviewer 3 Report

The article by Parisi and colleagues presents a deep mechanistic insights of CFTR in cancers. The review article is well written for the general audience to dwell into the CFTR and to get the updated information in this field. I recommend this review to be published in Cancers after taking care of following comments.

Authors should present a schematic showing functions of CFTR in various types of cancers.

Include CFTR Gene and protein structure for better clarity to the readers.

Discuss classes of CFTR mutations as a separate section.

Also, discuss regarding the CFTR modulators that are being currently used.

Author Response

I am very happy that our work has interested many reviewers and I would be happy to respond to this third reviewer as well.

1) Authors should present a schematic showing functions of CFTR in various types of cancers.

This is already shown in Figure 1 and Table 1.

2) COMMENTS:
 - Include CFTR Gene and protein structure for better clarity to the readers.
 - Discuss classes of CFTR mutations as a separate section.
 - Also, discuss regarding the CFTR modulators that are being currently used.

These are certainly interesting topics. However, they are out of place in such a review because they concern pathology in general and not the focus of the article. 

Round 2

Reviewer 2 Report

The manuscript has been improved, as requested. No further comments